

# Xenon ameliorates chronic post-surgical pain by regulating mitophagy in microglia and rats mediated by PINK1/Parkin pathway

Hu Lv[*], Jiaojiao Huang[*], Xin Zhang, Zhiyong He, Jun Zhang and Wei Chen

[1] Department of Anesthesiology, Fudan University Shanghai Cancer Center, Shanghai, China
[2] Department of Oncology, Shanghai Medical College, Fudan University, Shanghai, China
[*] These authors contributed equally to this work.

## ABSTRACT

**Background.** Chronic post-surgical pain (CPSP) is one of the important causes of poor postoperative outcomes, the activation of microglia in the spinal cord is closely related to the generation, transmission and maintenance of CPSP. Xenon (Xe), an anesthetic gas, has been reported to be able to significantly reduce intraoperative analgesia and postoperative pain sensation at low doses. However, the mechanism of the regulatory effect of xenon on activated microglia after CPSP remains unclear.

**Methods.** In this study, CPSP model rats were treated with 50% Xe inhalation for 1 h following skin/muscle incision and retraction (SMIR), once a day for 5 consecutive days, and then the painbehavioraltests (pain behavior indexes paw withdrawal mechanical threshold, PWMT and thermal withdrawal latency, TWL), microglial activation, oxidative stress-related indexes (malondialdehyde, MDA; superoxide dismutase, SOD; hydrogen peroxide, $H_2O_2$; and catalase, CAT), mitophagy and PINK1/Parkin pathway were examined.

**Results.** The present results showed that a single dose of Xe treatment in SMIR rat model could significantly improve PWMT and TWL in the short-term at a single treatment and long-term at multiple treatments. Xe treatment inhibited microglia activation and oxidative stress in the spinal dorsal horn of SMIR rats, as indicated by the decrease of Iba1 and MDA/$H_2O_2$ levels and the increase of SOD/CAT levels. Compared with the control group, Xe further increased the CPSP promoted Mito-Tracker (a mitochondrial marker) and LC3 (an autophagy marker) co-localization positive spots and PINK1/Parkin/ATG5/BECN1 (autophagy-related proteins) protein expression levels, and inhibited the Mito-SOX (a mitochondrial reactive oxygen species marker) positive signal, indicating that Xe promoted microglia mitophagy and inhibited oxidative stress in CPSP. Mechanistically, we verified that Xe promoted PINK1/Parkin signaling pathway activation.

**Conclusion.** Xe plays a role in ameliorating chronic post-surgical pain by regulating the PINK1/Parkin pathway mediated microglial mitophagy and provide new ideas and targets for the prevention and treatment of CPSP.

Corresponding authors
Jun Zhang, snapzhang@aliyun.com
Wei Chen,
chenwei_fdu@fudan.edu.cn

## INTRODUCTION

Chronic post-surgical pain (CPSP) occurs after surgery that lasts at least more 3 months, excluding pain caused by malignant tumors, chronic infections and other factors (*Glare, Aubrey & Myles, 2019*). Approximately 10–50% of patients experience persistent pain after surgery, and as the duration of severe postoperative pain increases, the incidence of chronic postoperative pain increases significantly (*Humble, Dalton & Li, 2015*). Pain is related to the surgical area but may also involve different areas and has a neuropathic component (*Li et al., 2021*). The occurrence of CPSP can lead to adverse consequences and seriously affect the recovery and quality of life of patients (*Tawfic et al., 2017*). Due to the high incidence of CPSP, the specific pathogenesis of it is still unclear, so there is no rapid and effective prevention and treatment.

The inflammatory process after surgery is one of the factors contributing to persistent pain (*Esses et al., 2020*). Growing evidence suggests that peripheral and central nervous system inflammation is closely related to chronic pain (*Vergne-Salle & Bertin, 2021*; *Walker et al., 2014*). One of the hallmarks of neuroinflammation is the activation of microglia, leading to the release of pro-inflammatory factors (*Ji et al., 2018*). Previous studies have found that increased levels of oxidative stress due to mitochondrial dysfunction are strongly associated with chronic neuropathic pain (*Lv et al., 2016*). Mitophagy is a way of maintaining cellular homeostasis, and when mitochondrial function is impaired, it can lead to more severe cellular inflammation. Studies have shown that abnormal autophagy function of microglia is involved in the pathogenesis of neuropathic pain (*Liu et al., 2019*). Induction of mitophagy can inhibit neuroinflammation and exert neuroprotective effects through various pathways (*Lautrup et al., 2019*; *Lou et al., 2020*).

Xenon (Xe), an inert gas, is considered to be a promising volatile anesthetic and has been successfully used in the fields of gynecology, urology, ophthalmology, and cardiac surgery (*Nair et al., 2021*). In addition to its anesthetic effect, it also has neuroprotective, anti-stress, immune stimulation and analgesic effects (*Maze & Laitio, 2020*; *Le Nogue et al., 2020*). It has been confirmed that the analgesic effect of xenon is explained by its inhibitory effect on NMDA receptors in nociceptive systems (*Georgiev et al., 2010*). Xenon is nontoxic, nonirritant, and harmless to the body because it does not undergo metabolism and biotransformation in the body, which allows the use of this gas in the treatment of severe pain syndromes in oncology and neurology (*Zhang et al., 2021*). However, there are no studies on the efficacy of xenon for CPSP.

Based on the above research basis, this study aims to further study the protective effect and mechanism of Xe on CPSP. CPSP model was established by skin/muscle incision and retraction (SMIR) and then 50% xenon (Xe) inhalation therapy was administered for 1 h. Through a series of experiments, the efficacy and therapeutic mechanism of Xe on CPSP were comprehensively evaluated from animal and cellular levels. It will provide experimental basis for the pathogenesis and clinical treatment of CPSP, and provide more theoretical support for the clinical application of Xenon.

## MATERIALS & METHODS

### SMIR model construction and Xe treatment

Eighty healthy adult male Sprague-Dawley (SD) rats, 6–8 weeks, 230–300 g, were purchased from Shanghai Laboratory Animal Research Center. Afterwards, they were reared in an alternating day and night environment with free food and water. All animal-related experiments were reviewed and approved by the Fudan University Shanghai Cancer Center Ethics Committee (FUSCC-IACUC-S2022-0346).

The SMIR surgery was performed as previously reported (*Pan et al., 2019*), shown in Fig. S1. In brief, 60 SD rats were anesthetized with 40 mg/kg 1% sodium phenobarbital (Sinopharm, China) intraperitoneally, and then fixed on the operating table in a supine position. The inner thigh of one side was shaved and then disinfected with alcohol to expose the saphenous vein. A skin incision of 1.5–2 cm is made about four mm medial to the saphenous vein in the middle of the thigh to expose the muscle. A 7–10 mm incision is made in the superficial muscle four mm medial to the saphenous nerve. The superficial muscles were bluntly separated with scissors, and a retractor was placed to retract the skin and superficial muscles to two cm, and exposed the underlying adductor fascia. After 1 h, the skin and muscles were sutured. During the stretch, the rats were given sterile saline gauze to cover the incision for moisturizing and keeping warm.

SMIR Rats were divided into four groups, six rats in each group. In the sham operation group, the rest of the operation procedures were the same except that the skin and muscles were not stretched. Xe group rats received 50% Xe (Wuhan Newread Special Gas Co., Ltd, China) inhalation once for 1 h. CPSP group rats received SMIR surgery. CPSP + Xe group received SMIR operation and 50% Xe inhalation for 1 h, once a day for 5 consecutive days.

### Behavioral experiment

#### Paw withdrawal mechanical threshold (PWMT) detection

An electronic automatic paw tactile tester (North Coast Medical Inc., Morgan Hill, CA, USA) was used to respectively detect the PWMT of the hindlimbs on the operative and non-operative sides within 0–5 h and 2–14 d after Xe treatment, as previously described (*Chaplan et al., 1994*). Baseline values were measured 1 day before surgery. Rats were individually placed in a plexiglass frame (28 × 22 × 18 cm) with a metal mesh pad at the bottom and allowed to acclimate for at least 30 min before testing. Using a commercial electronic von Frey instrument, pressure was applied to the mid-plantar surface of the right hind foot with von Frey fibers. When the rats developed pain response (sudden paw withdrawal, shaking or hind paw licking) due to von Frey's pressure, the numbers on the computer display were recorded. Tests were spaced at least 5 min apart to eliminate interference from previous painful stimuli. Three consecutive test readings were collected, and the average of the three test readings was used as a paw withdrawal mechanical threshold (PMWT) value. To minimize animal discomfort, mechanical nociceptive thresholds were tested on only one hind paw.

## Thermal withdrawal latency (TWL) detection

The basal value was detected 1 day before surgery using a paw heat pain tester (Shanghai Yuyan Scientific Instrument Co Ltd., Shanghai, China). TWL was detected on the injured side and the unaffected side within 0–5 h and 2–14 d after Xe treatment based on a design by *Hargreaves et al. (1988)*. The plantar stimulation analgesia was used for detection, the temperature-regulated glass platform was kept at 29 °C, and the animals were acclimated on the platform for 10–15 min. A radiant heat source was placed on the hind sole, and the thermal contraction latency was recorded. Thermal paw withdrawal latency, the time it takes for a rat to remove its hind paw from a heat source, is used to assess thermal hyperalgesia. Left and right paws were tested at random intervals of 1 min. The two posterior grasps were tested 3 times, with a 5-minute interval between the two tests. Data is expressed in seconds.

## ELISA kit test

The spinal cord dorsal horn tissue of each group of animals was collected 14 days after operation, and the supernatant was taken after the tissue was homogenized. The oxidative stress indicators malondialdehyde (MDA) superoxide dismutase (SOD), hydrogen peroxide ($H_2O_2$) and catalase (CAT) in each group were detected according to the instructions of the ELISA kit (Shanghai Enzyme-linked Biotechnology Co., Ltd., Shanghai, China). The specific operations are as follows: Add sample/standard and incubate at 37 °C for 1 h. After washing three times, add enzyme conjugate at 37 °C for 30 min. After washing three times, add chromogenic solution and incubate at 37 °C for 30 min. The reaction was terminated by adding stop solution. Then the absorbance value was detected at 450 nm.

## Western blot

Rats were anesthetized with 40 mg/kg 1% sodium phenobarbital intraperitoneally. The spinal cord tissue was quickly removed, lysed with RIPA lysis buffer (Bio-Rad Laboratories, Inc., Hercules, CA, USA) containing 1mM phenylmethanesulfonyl fluoride (PMSF, Thermo Fisher Scientific, Waltham, MA, USA), and homogenized by ultrasound on ice. RIPA lysis buffer was added to cell sample and lysed on ice for 30 min. Centrifuge at 12,000 rpm for 15 min at 4 °C, and take the supernatant. Protein electrophoresis was performed after protein quantification by BCA kit (Thermo Fisher Scientific, Waltham, MA, USA). The total protein was separated by SDS-PAGE gel (MCE, USA.) and transferred to PVDF membrane (Thermo Fisher Scientific) Then the PVDF membrane was blocked with 5% w/v BSA (Sigma-Aldrich, St. Louis, MO, USA) for 1 h at room temperature. After washing three times, add anti-Iba1 antibody (1:1000, Abcam, Cambridge, UK), anti-PINK1 antibody (1:1000, Abcam, Cambridge, UK), anti-Parkin antibody (1:1000, Abcam, Cambridge, UK), anti-ATG5 antibody (1:1000, Abcam, Cambridge, UK), anti-BECN1 antibody (1:1000, Abcam, Cambridge, UK), anti-actin antibody (1:1000, Abcam, Cambridge, UK) and incubate at 4 °C overnight. After washing 3 times, add horseradish peroxidase (HRP)-labeled secondary antibody (Abcam, Cambridge, UK) and incubate at room temperature for 1 h. Finally, add ECL solution (Thermo Fisher Scientific) to exposure for 1∼3 min. Protein bands were analyzed using Image J and optical density was detected. $\beta$-actin was used as an internal reference.

## Immunofluorescence histochemical staining

Rats were anesthetized with 40 mg/kg 1% sodium phenobarbital intraperitoneally. After laparotomy, normal saline was rapidly perfused through the aorta, followed by 4% paraformaldehyde (Solarbio, Beijing, China) for 30 min. Spinal cords were removed and fixed in 4% paraformaldehyde for 3 h, then transferred to 30% sucrose (Sigma-Aldrich, USA) for 2 days of dehydration. The slices were cut to 25 $\mu$m thick under freezing conditions. Spinal cord sections were rinsed 3 times with PBS (GIBCO, Invitrogen Corp, Carlsbad, CA, USA), 5 min each time, and blocked for 1 h at room temperature. Anti-Iba1 antibody (Abcam, Cambridge, UK) was added, overnight at 4 °C. Remove the anti-Iba1 antibody, rinse with PBS for 3 times, add fluorescently labeled secondary antibody (Abcam, Cambridge, UK) for 1 h at room temperature in the dark. The sections were rinsed 3 times with PBS and observed under a fluorescence microscope and photographed.

## Microglia extraction

The extraction of microglia refers to the method of Pacheco (*Pacheco et al., 2020*), and the specific operations are as follows: Take three 1–2 days old neonatal SD rats, quickly cut the skull under sterile conditions, take out the cerebral hemisphere tissue into a petri dish containing HBSS (GIBCO, Invitrogen Corp, Carlsbad, CA, USA) + 10% fetal bovine serum (FBS, GIBCO, Invitrogen Corp, Carlsbad, CA, USA) solution. Then carefully peel off the meninges under the microscope. The brain tissue was cut into small pieces, placed in a centrifuge tube containing HBSS+10% FBS solution, and placed on ice for 1 h. The brain tissue was washed 3 times with three mL of HBSS + 10% FBS solution and HBSS in turn. Add three mL of 0.25% trypsin (GIBCO, Invitrogen Corp, Carlsbad, CA, USA) solution and place in a water bath at 37 °C for 30 min. The brain tissue was washed 3 times with three mL of HBSS + 10% FBS solution and HBSS successively. Add seven mL of HBSS and centrifuge at 450 × g for 5 min at 4 °C. Discard the supernatant and add four mL of HBSS + 10% FBS to resuspend the pellet. Transfer the cells to a T75 culture flask, add 10 mL DMEM (Gibco, Waltham, MA, USA) with 10% FBS and continue to culture in a 5% $CO_2$, 37 °C cell incubator. On third and fifth day, the medium was replaced with DMEM medium containing 10% FBS. By around day 14, microglia were confluent for purification. Fix the culture flask on a constant temperature shaker at 37 °C and shake (180 rpm) for 1–2 h. The medium containing microglia was collected, centrifuged at 4 °C (800 rpm, 8 min), the supernatant was discarded, and the cells were resuspended in DMEM medium containing 10% FBS to continue the culture.

## Construction of CPSP microglia cell model

The CPSP microglia cell model was constructed by collecting the cerebrospinal fluid of CPSP rats and co-cultured with primary microglia for 24 h.

The cells were divided into four groups, the cells in the control group were primary microglia cells, the cells in the Xe group were treated with Xe for 1 h, the cells in the CPSP group were CPSP microglia, and the cells in the CPSP + Xe group were CPSP microglia treated with Xe for 1 h.

## Mito-SOX fluorescent staining

Add MitoSOX™ Red (1:1000; Invitrogen, Waltham, MA, USA) to cells at 37 °C for 30 min. After washing with PBS for 3 times, observed with a fluorescence microscope and photographed (OLYMPUS, Tokyo, Japan).

## Co-localization of GFP-LC3 and Mito Tracker fluorescent staining

The cells were added with Mito-Tracker Red CMXRos (1:1000, Invitrogen) 37 °C for 30 min. After washing with PBS for three times, add anti-LC3 antibody overnight at 4 °C. Then add fluorescently labeled secondary antibody (Abcam), for 1 h at room temperature in the dark. The cells were rinsed three times with PBS and observed under fluorescence microscope and photographed.

## SiRNA design, transfection and screening

Two double-stranded siRNA targeting rat PINK1 mRNA (5′-TGAAGAGACTCAGGCGCTA-3′, 5′-GCGAGGTGGTGAAGAGACT-3′) were designed using siRNA Target Finder and Design software (http://www.genelink.com/sirna/RNAicustomorder.asp) and synthesized by Gemma Gene Company. The control group was Scramble siRNA with no homology.

According to the instructions, Lipofectamine 2000 (Invitrogen) was used to transfect different PINK1 siRNAs into microglia. The transfection experiment was carried out when the cells had grown to more than 80%. First, 20 µL of siRNA was mixed with 250 µL of Opti-DMEM (Invitrogen), and then placed at room temperature for 5 min. Then 10 µL of Lipofectamine 2000 was mixed with 250 µL of Opti-DMEM, and place at room temperature for 5 min. The two solutions were mixed well and left at room temperature for 15 min. Add the mixture to the cell plate and shake the cell plate gently. After transferring the cell plate to the incubator for 6–8 h, replace with new DMEM medium and continue to cultivate for 72 h.

The effectiveness of siRNA was detected by Western blot. Validated PINK1 siRNA were selected for subsequent *in vivo* experiments.

## Statistical analysis

SPSS 16.0 (SPSS Inc., Chicago, IL, USA) and GraphPad Prism 8.3.0 (GraphPad Software, La Jolla, CA, USA) statistical analysis software were used for data analysis and processing of experimental results. Measurement data were expressed as mean ± standard deviation (mean ± SD). The independent sample $t$-test was used to analyze the data between the two groups, and the one-way ANOVA test was used to analyze the pairwise data of three or more groups. $P < 0.05$ was considered statistically significant for differences.

# RESULTS

## Xe improved PWMT and TWL in CPSP rat model

The effect of Xe on pain improvement following SMIR surgery was tested in the rat CPSP model (Fig. S1). The rats were established by SMIR surgery, and then Xe was inhaled for 1 h (50%) immediately, for 5 consecutive days. The PWMT and TWL were measured at BL (baseline), 0, 1, 2, 3, 4, 5 h (short-term) and day 2, 4, 6, 8, 10, 12 and 14 (long-term)
after CPSP using the von Frey test and paw heat pain tester. The specific timeline of the individual experiment was shown in Fig. 1A. Behavioral testing showed that PWMT was significantly decreased after CPSP compared with the sham control. After xenon treatment, PWMT increased, but there was no significant difference compared with CPSP group 2 h after xenon treatment (Fig. 1B). Meanwhile, TWL was also significantly increased after CPSP, but no statistical difference after 5 h with xenon treatment compared with CPSP group (Fig. 1C). This result demonstrated that the Xe may be effective in relieving pain caused by SMIR in the short-term at a single treatment. PWMT and TWL were further measured at days 2, 4, 6, 8, 10, 12, and 14. Figures 1D and 1E shown that PWMT and TWL were all significantly increased at these time points compared with CPSP group, revealed that Xe also may be effective in relieving pain caused by SMIR in the long-term at multiple treatments.

### Xe repressed oxidative stress and microglial activation in the dorsal horn of rat CPSP model

Considering that oxidative stress plays a critical role in the peripheral nervous system changes in both acute inflammatory pain and chronic neuropathic pain (*Shim et al., 2019*), we investigated whether oxidative stress was also activated after CPSP and whether Xe treatment was associated with oxidative stress. Spinal cord dorsal horn tissues of each group were collected 14 days after SMIR surgery and Xe treatment, and oxidative stress-related markers (MDA/SOD/$H_2O_2$/CAT) were detected by ELISA kits. The levels of MDA and $H_2O_2$ level were significantly increased in the CPSP group, accompanied by a significant decrease in SOD/CAT levels, while Xe inhalation significantly reduced CPSP-induced oxidative stress changes (Figs. 2A–2D). These results demonstrated that oxidative stress was observably activated after CPSP, which was further effectively inhibited by Xe treatment.

Microglia are a type of glial cells found throughout the brain and spinal cord (*Yan et al., 2022*). Microglial activation changes in morphology and phenotype occur after cerebral ischemic injury, and the severity of cerebral ischemic injury is related to the activation state of microglia (*Var et al., 2021*). To investigate whether Xe inhibited the activation of microglia in CPSP rats, microglia after Xe treatment were analyzed by Western Blot and immunofluorescence analysis. The Western blot analysis results (Figs. 2G and 2H) indicated that the protein expression level of Iba1 (microglial marker) in the spinal dorsal horn of rats induced by SMIR was significantly up-regulated, while its protein expression was observably down-regulated after Xe treatment. Moreover, Xe inhibited SMIR-induced microglial activation as measured by the intensive ramified Iba1-positive staining (Figs. 2E and 2F). These results suggested that Xe has an important inhibitory effect on oxidative stress and microglial activation in the rat CPSP model.

### Xe promoted microglia mitophagy and inhibited oxidative stress *in vitro*

To explore the role of Xe in activating microglia *in vitro*, we studied the viability of rat primary microglia incubated with CPSP rat cerebrospinal fluid in the presence or absence of Xe treatment for 0.5 h. Firstly, the content of Mito-SOX, a specific marker

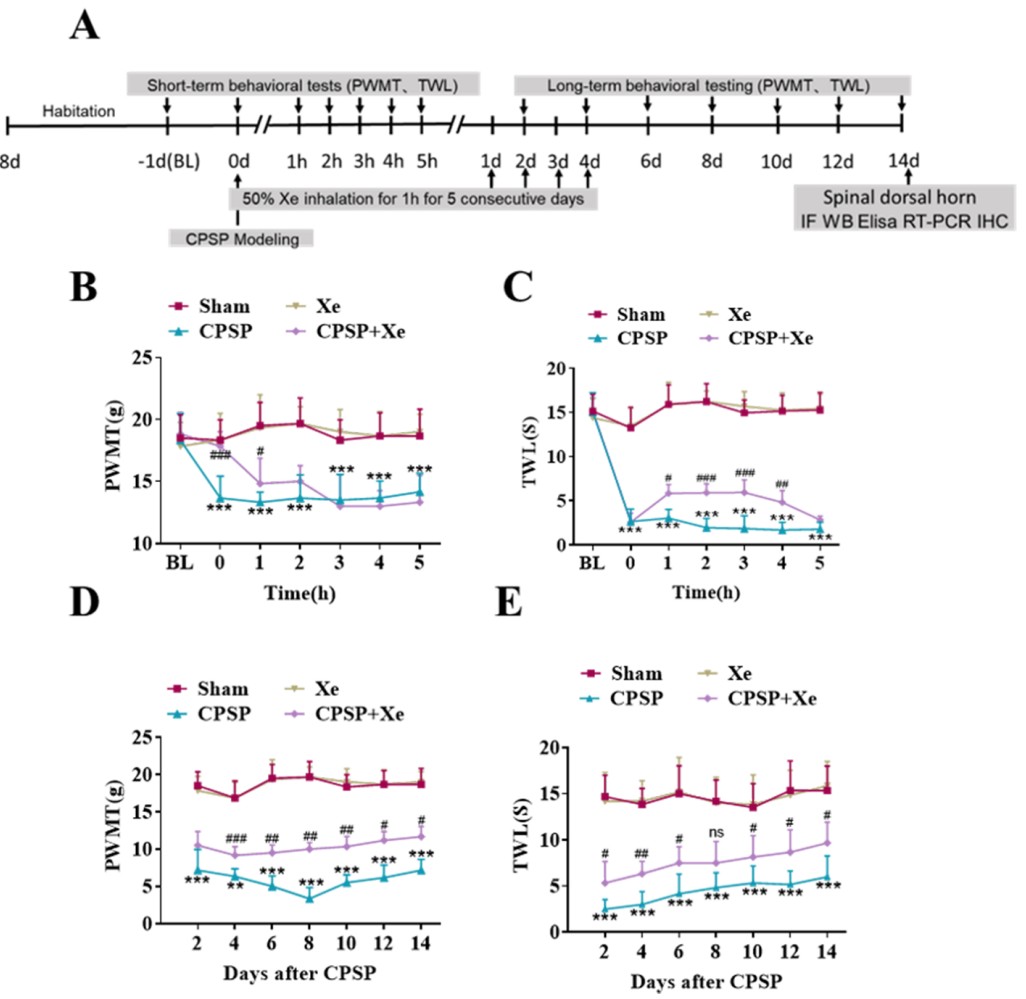

**Figure 1 Xe improved PWMT and TWL in CPSP rat model.** (A) Schematic illustration of experimental grouping and Xe treatment scheme. (B and C) Changes of PWMT (B) and TWL (C) at BL and hour 0, 1, 2, 3, 4, 5 after SMIR surgery following by Xe inhalation therapy (50%) for 1 h. (D and E) Changes of PWMT (D) and TWL (E) at days 2, 4, 6, 8, 10, 12 and 14 after SMIR surgery following by Xe inhalation therapy (50%) for 1 h, for 5 consecutive days. $n = 12$ of each group. BL, baseline . $*P < 0.05$, $**P < 0.01$, $***P < 0.001$ CPSP group *vs* sham group. $\#P < 0.05$, $\#\#P < 0.01$, $\#\#\#P < 0.001$ CPSP+Xe group *vs* CPSP group.

of mitochondrial reactive oxygen species, was detected by Mito-SOX fluorescent probe under inverted fluorescence microscope in each group. Compared with the control group, the Mito-SOX positive signal (red) of microglia in the CPSP model cells was significantly increased. However, the positive signal of Mito-SOX was significantly reduced after Xe treatment (Figs. 3A and 3B). Then, co-localization (yellow) of GFP-LC3 (green) and Mito Tracker (red) was used to detect the occurrence of mitophagy. The staining results showed that compared with the control group, the CPSP group was significantly increased the co-localization; It is worth noting that Xe treatment further significantly enhanced CPSP-induced the co-localization (Fig. 4B). Finally, the expression

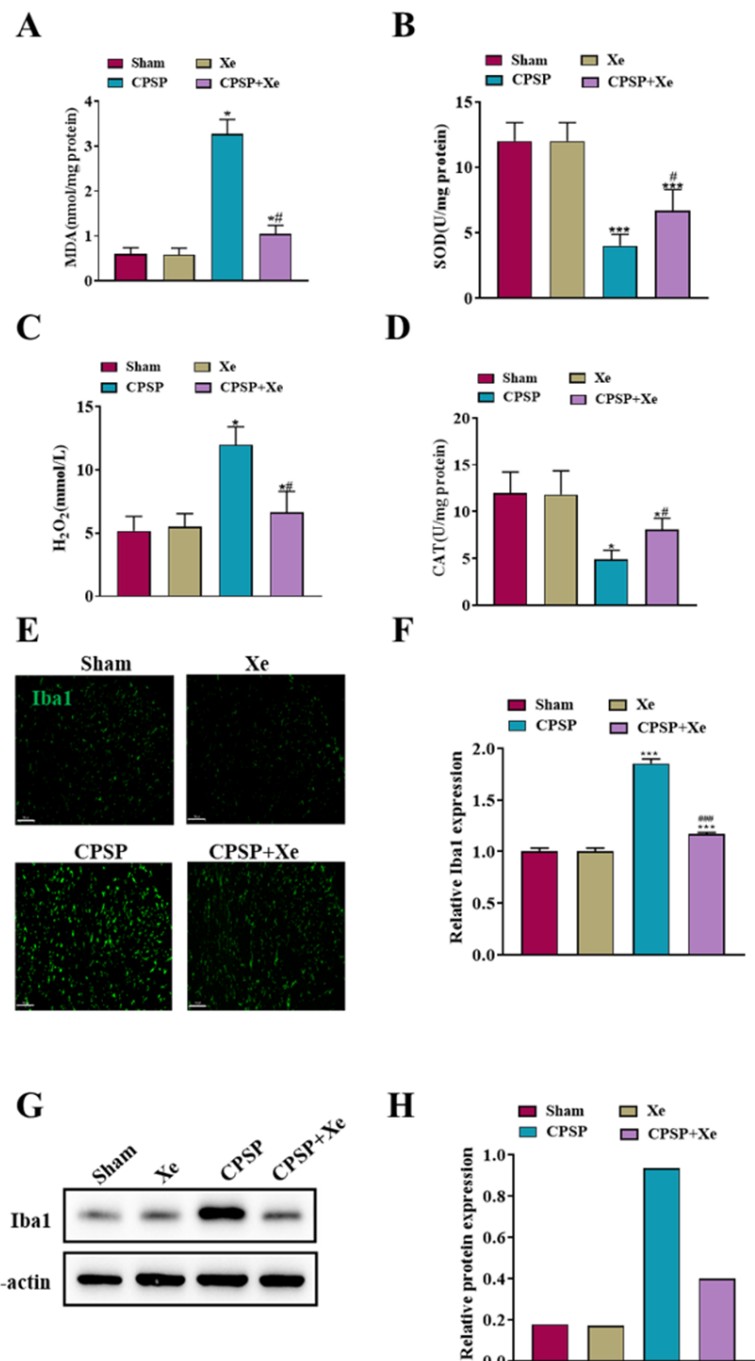

**Figure 2** **Xe repressed oxidative stress and microglial activation in the dorsal horn of rat CPSP model.**
(A–D) MDA (A), SOD (B), $H_2O_2$ (C), and CAT (D) content in spinal cord dorsal horn of CPSP rats were detected by ELISA analysis. (E) Tissue immunofluorescence staining images showing the expression of Iba1 in the spinal dorsal horn of CPSP rats with Xe treatment. ('200, scale bar = 100 μm). (F) Quantitative statistics of Iba-1 intensity. (G) Western blot detection of Iba1 protein expression levels in the spinal dorsal horn of CPSP rats with Xe treatment. (H) Quantitative statistics of Iba1 protein expression. *$P <$ 0.05, ***$P <$ 0.001 $vs$ sham group. #$P <$ 0.05, ###$P <$ 0.001 $vs$ CPSP group.

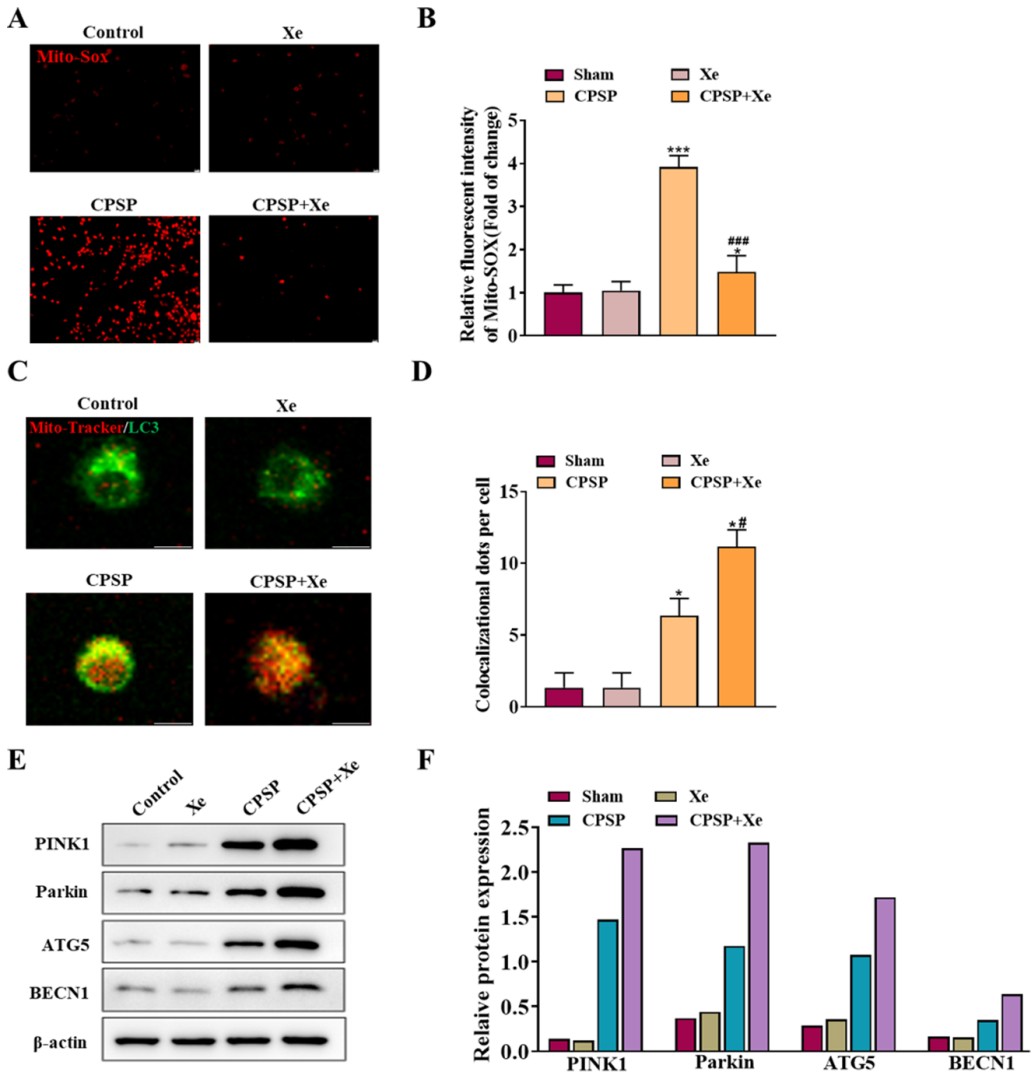

**Figure 3** **Xe promoted microglia mitophagy and inhibited oxidative stress *in vitro*.** (A) Effect of Xe (50%) on oxidative stress levels in CPSP model cells was assessed by Mito-SOX fluorescent probe under inverted fluorescence microscope. (×100, scale bar = 20 μm). (B) Relative fluorescent intensity of Mitro-SOX. (C) Fluorescence staining to detect the co-localization of Mito-Tracker and LC3 in CPSP model cells and Xe treatment. (×400, scale bar = 10 μm). (D) Co-localization of Mito-Tracker and LC3 dots per cells. (E) Western blot of PINK1/Parkin/ATG5/BECN1 protein expression in CPSP model cells and Xe treatment. (F) Quantitative statistics of PINK1/Parkin/ATG5/BECN1 protein expression. *$P < 0.05$, ***$P < 0.001$ *vs* control group. #$P < 0.05$, ###$P < 0.001$ *vs* CPSP group.

of autophagy-related protein PINK1/Parkin/ATG5/BECN1 was detected by Western blot analysis. The results showed that compared with the control group, the protein expression levels of PINK1/Parkin/ATG5/BECN1 in the CPSP group were significantly increased, and Xe treatment also significantly increased the expression of autophagy-related proteins (Fig. 3C). These results demonstrated that Xe treatment promotes microglia autophagy and may mitigate mitochondrial dysfunction, thereby reducing oxidative stress in microglia.

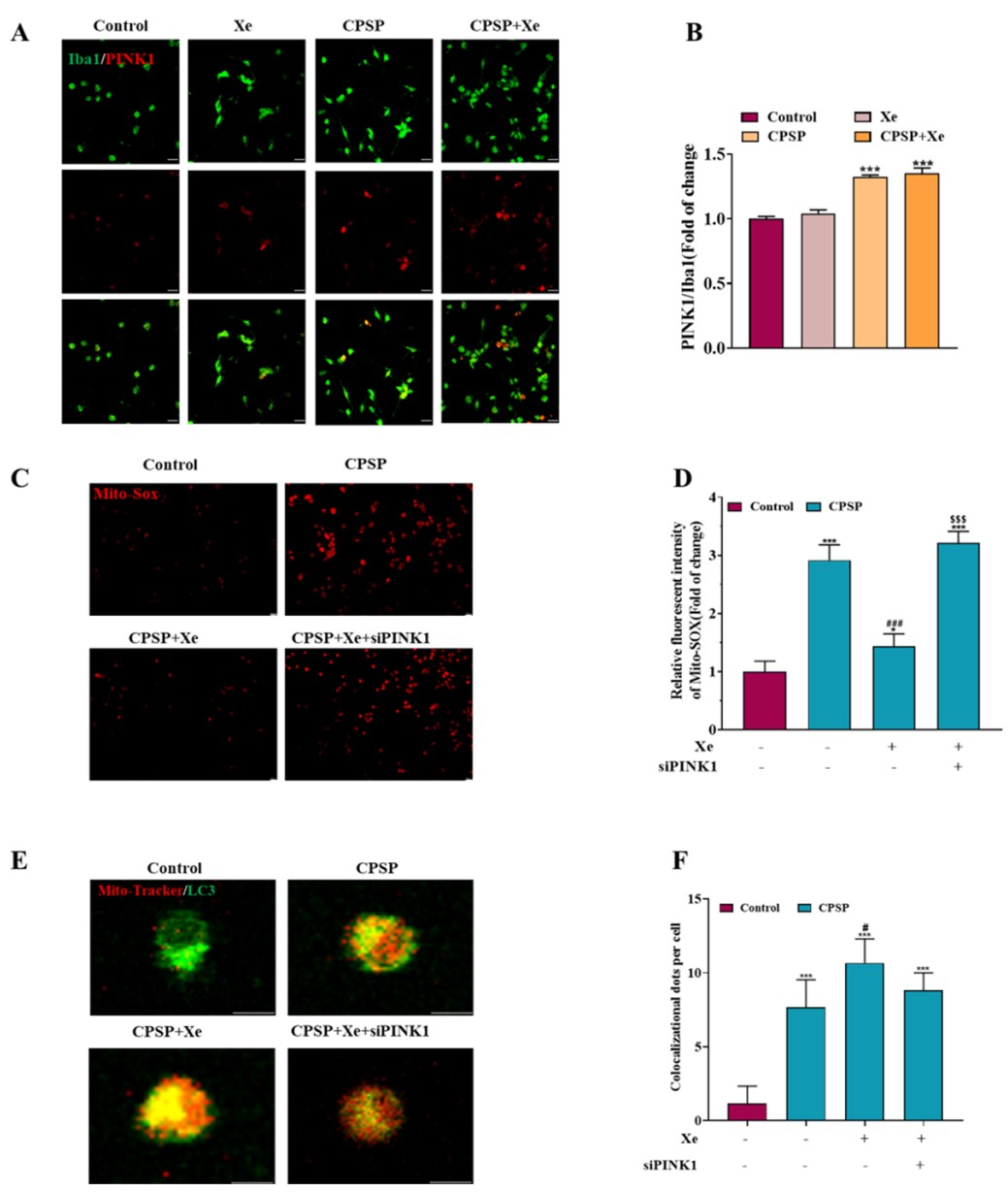

**Figure 4  Xe regulated mitophagy by activating the PINK1/Parkin pathway.** (A) Double immunofluorescence staining shows that PINK1 were mainly co-localized with Iba1. (×200, scale bar = 20 μm). (B) Quantitative statistics of PINK1/ Iba1 co-localization. (C). Effect of PINK1 knockdown on oxidative stress levels in CPSP model cells was assessed by Mito-SOX fluorescent probe under inverted fluorescence microscope. ('100, scale bar = 20 μm) (D) Relative fluorescent intensity of Mitro-SOX. (E) Fluorescence staining to detect the co-localization of Mito-Tracker and LC3 in CPSP model cells and PINK1 knockdown. (×400, scale bar = 10 μm). (F). Co-localization of Mito-Tracker and LC3 dots per cells. $n = 3$. *$P < 0.05$, **$P < 0.01$, ***$P < 0.001$ *vs* control group. #$P < 0.05$, ###$P < 0.001$ *vs* CPSP group. $P < 0.001$ *vs* CPSP+Xe group.

## Xe regulated mitophagy by activating the PINK1/Parkin pathway

Studies have shown that PINK1-mediated mitophagy has been implicated in neuropathic pain (*Dionisio et al., 2019*). Therefore, we investigated whether Xe induced activation of PINK1/Parkin signaling pathway in a rat CPSP cell model. Firstly, the level of PINK1 in Iba1-positive microglia in the CPSP group was detected by immunofluorescence, and it was found that the level of PINK1 (red) in Iba1-positive cells (green) in the CPSP group was significantly higher than that in the control group, whereas Xe treatment significantly increased PINK1 levels (Figs. 4A and 4B). This result demonstrated that Xe activates the PINK1/Parkin pathway. In addition, we constructed siRNA (siPINK1-1 and siPINK1-2) to knockdown PINK1 in CPSP model cells. The effect of siRNA transfection was verified by Western blot analysis, and siPINK1-1 with the most obvious effect was used as a research tool for subsequent experiments (Figs. S2A and S2B). Then, the effect of Xe on the mitochondrial oxidative stress marker Mito-SOX after knockdown of PINK1 in CPSP model cells was detected. The staining results showed that the positive signal of Mito-SOX was significantly reduced after Xe treatment, while the positive signal was further increased after knockdown of PINK1 (Figs. 4C and 4D). Finally, the co-localization of GFP-LC3 and Mito Tracker was reduced in CPSP model cells after Xe treatment, whereas it was significantly increased after PINK1 knockdown (Figs. 4E and 4F). The above results suggested that Xe can promote microglia mitophagy and inhibited oxidative stress in CPSP by activating the PINK1/Parkin pathway.

## Inhibition of PINK1 disrupted the improvement of PWMT and TWL in the rat CPSP model with Xe treatment

The effect of Xe on the pain behavior of CPSP rats after Pink1 knockdown was further studied at the animal level. Intrathecal injection of siPINK1 interfered with lentivirus 48 h before CPSP was used to study the mechanism of Xe therapy (Fig. 5A). Immunofluorescence analysis results (Figs. 5B and 5C) indicated that siPINK1 promoted microglial activation that was inhibited by Xe treatment in the spinal dorsal horn of CPSP rats as measured by the intensive ramified Iba1-positive staining. In addition, the protein expression level of Iba1 was also up-regulated in in the spinal dorsal horn of CPSP rats with both siPINK1 injection and Xe treatment (Figs. 5D and 5E). PWMT was significantly reduced after siPINK1 injection, but no statistical difference after 2 h compared with CPSP group after Xe treatment (Fig. 5F). Meanwhile, TWL was also significantly reduced after siPINK1 injection, but no statistical difference after 5 h compared with CPSP group after Xe treatment (Fig. 5G). PWMT and TWL were further measured at days 2, 4, 6, 8, 10, 12, and 14 (Figs. 5H and 5I). As shown that PWMT and TWL were all significantly reduced at these time points compared with CPSP group after Xe treatment. These results suggested that siPINK1 disrupted the improvement of PWMT and TWL in the rat CPSP model with Xe treatment.

## DISCUSSION

In this study, we investigated the potential role of Xe in relieving CPSP, and identified the effect of Xe on regulating mitophagy in the rat CPSP model. The present data verified

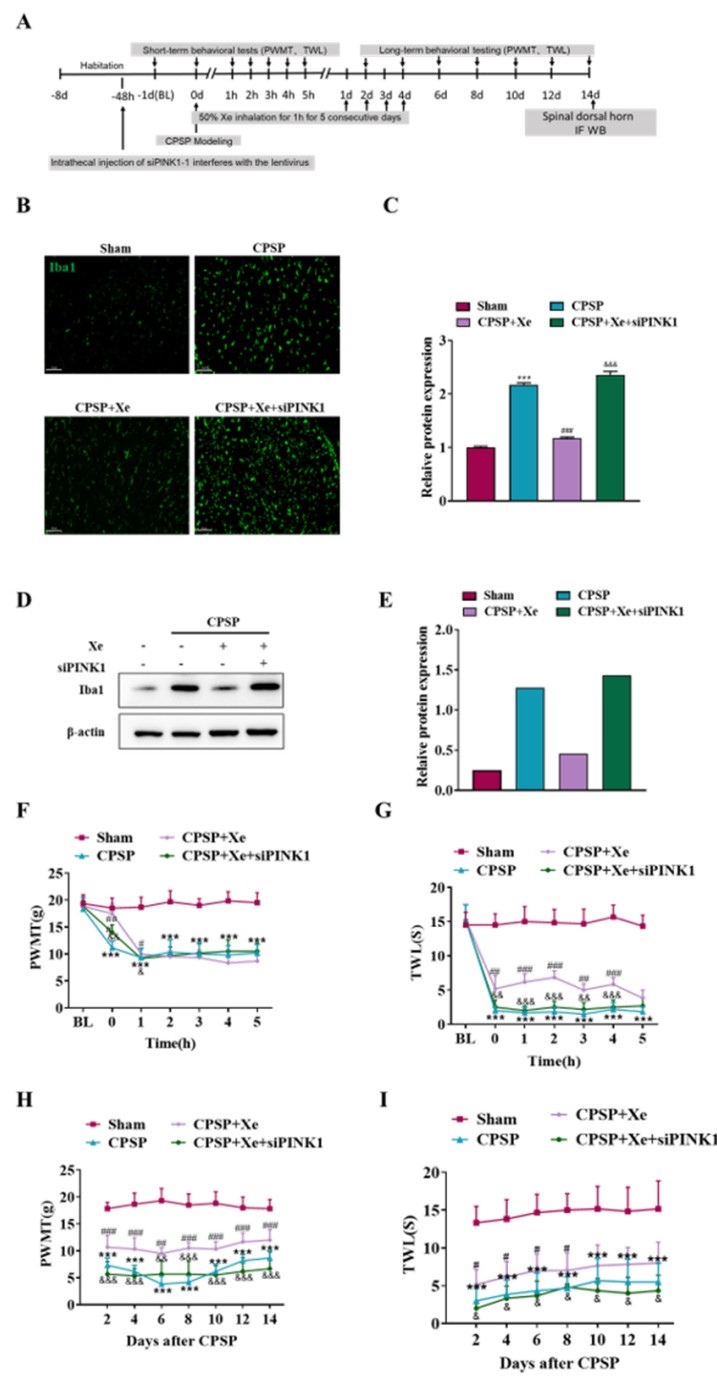

**Figure 5   Inhibition of PINK1 disrupted the improvement of PWMT and TWL in the rat CPSP model with Xe treatment.** (A) Experimental grouping of intrathecal injection of PINK1 and schematic diagram of Xe treatment scheme. (B) Tissue immunofluorescence staining images showing the expression of Iba1 in the spinal dorsal horn of CPSP rats with Xe treatment and injection siPINK1 through the spinal cord. ($\times$200, scale bar $=$ 100 $\mu$m). (C) Quantitative statistics of Iba-1 intensity. (D) Western blot detection of Iba1 protein expression levels in the spinal dorsal horn of CPSP rats with Xe treatment and injection siPINK1 through the spinal cord. (E) Quantitative statistics of Iba1 protein expression. (F and G) Changes

**Figure 5 (...continued)**
of PWMT (F) and TWL (G) at BL and hour 0, 1, 2, 3, 4, 5 after SMIR surgery following by Xe inhalation therapy (50%) for 1 h and injection siPINK1 through the spinal cord. (H and I) Changes of PWMT (H) and TWL (I) at days 2, 4, 6, 8, 10, 12 and 14 after SMIR surgery following by Xe treatment for 1 h injection siPINK1 through the spinal cord. $n = 6$ of each group. $*P < 0.05$, $**P < 0.01$, $***P < 0.001$ CPSP group $vs$ Sham control group. $\#P < 0.05$, $\#\#P < 0.01$, $\#\#\#P < 0.001$ $vs$ CPSP+ Xe group $vs$ CPSP group. $\& P < 0.05$, $\&\& P < 0.01$, $\&\&\& P < 0.001$ CPSP+Xe+siPINK1 group $vs$ CPSP+Xe group.

that (1) Xe improved PWMT and TWL in the CPSP rat model; (2) Xe repressed oxidative stress and microglial activation in the dorsal horn of rat CPSP model; (3) Xe promoted microglia mitophagy and inhibited oxidative stress *in vitro*; (4) Xe regulated mitophagy by activating the PINK1/Parkin pathway; (5) Inhibition of PINK1 disrupted the improvement of PWMT and TWL in the rat CPSP model by Xe. These data validated the important role of Xe in activating the occurrence of mitophagy in microglia, which may provide a new opportunity for the treatment of CPSP.

Xe, an inert gas, has excellent anesthetic properties and can penetrate the blood–brain barrier without side effects. In the past decades, the protective effect of xenon on the central nervous system has been demonstrated (*Fries et al., 2008*). At present, researches only stay in the neuroprotective effect of xenon on nerve tissue pathology and improve nerve function, but there is no research on the effect of xenon on CPSP and the specific mechanism is still unclear. *Koziakova et al. (2019)* demonstrated that the neuroprotective effect of xenon may be mediated by inhibition of the *N*-methyl-d-aspartate receptor at the glycine site. *Kukushkin et al. (2017)* demonstrated that in simulated models of two kinds of physiological pain (formalin-induced acute pain and tonic pain), the TWL of pain behavior in rats was relieved after 15 min after inhalation of 50/50% xenon/oxygen for 30 min. The analgesic effect of PWMT was stable within 60 min (*Kukushkin et al., 2017*). Where xenon plays a major role in this sensitization is the long-lasting depolarization of glutamate and neurokinin exerted on nociceptive dorsal horn neurons (*Kukushkin & Igonkina, 2014*). Similar to their findings, our results showed that PWMT were improved significantly within 2 h after 1 h Xe (50%) inhalation treatment, while TWL improved significantly within 5 h. PWMT and TWL were also effectively restored after 1 h of Xe inhalation once daily for 5 consecutive days at days 2, 4, 6, 8, 10, 12, and 14 after SMIR surgery.

Notably, an increasing number of investigators have devoted to microglia and pain research in the past 10 years (*Han et al., 2021*; *Yang et al., 2017*; *Zhang et al., 2018*; *Liao et al., 2020*). *Liang et al. (2022)* demonstrated that microglia in spinal dorsal horn tissue were significantly activated after CPSP, and inhibiting the activation of microglia was one of the effective methods for the treatment of CPSP. Our results were consistent with that, and Xe inhalation treatment effectively inhibited microglial activation, but the specific mechanism of microglia function regulation during CPSP was still unclear.

Mitophagy is the selective degradation of mitochondria by autophagy, which usually occurs in defective mitochondria after damage or stress (*Shefa et al., 2019*). Mitophagy can be regulated by PINK1 and Parkin factors, and the PINK1/Parkin pathway is the most known mitochondrial autophagy system so far (*Yi et al., 2019*). PINK1, as a sensor

of mitochondrial polarization status, rapidly degraded in normal mitochondria. Impaired degradation of PINK1 in dysfunctional mitochondria leads to accumulation of PINK1 in the outer membrane of mitochondria, resulting in Parkin recruitment from the cytoplasm (*Jin et al., 2010*). Parkin on mitochondrial membrane can further recruit autophagy marker LC3 into mitochondria and promote mitochondrial autophagy (*Narendra et al., 2010*). *Piao et al. (2018)* found that the expression level of PINK1 was significantly up-regulated in the spinal cord dorsal horn after chronic constriction injury (CCI)-induced neuropathic pain mice model. In this study, PINK1 expression was significantly up-regulated in the spinal dorsal horn of CPSP, and further up-regulated after Xe treatment. Mito-Tracker and LC3 immunofluorescence staining showed that Xe could induce mitophagy after CPSP. We wanted to further explore whether the improvement effect of Xe on CPSP was realized by regulating PINK1/Parkin-mediated mitophagy. The current results showed that Xe promoted microglia mitophagy and inhibited oxidative stress in CPSP. PINK1 knockdown restored Xe-inhibited mitophagy and further increased mitochondrial oxidative stress. In addition, PINK1 knockdown abrogated the behavioral improvement of PWMT and TWL and the inhibitory effect on microglia activation in CPSP rats after Xe treatment.

There are still some shortcomings in this thesis: (1) lack of siPINK1 corresponding control siRNA NC group; (2) lack of mitochondrial function tests; (3) lack of experimental validation of whether Xe affects mitochondrial biogenesis; (4) lack of autophagy flux analysis experiments. We will further improve these shortcomings in the future.

In conclusion, we found that Xe had a restorative effect on pain behavior in a rat model of CPSP. Xe treatment helped to suppress microglial activation and oxidative stress. In addition, we demonstrated that Xe regulated mitophagy by activating PINK1/Parkin signaling. The current study confirmed that Xe has the effect of enhancing mitophagy and inhibiting oxidative stress in the rat model of CPSP, which may be a potential drug for the treatment of CPSP.

### Funding
This work was supported by the Natural Science Foundation of Shanghai (No. 21ZR1414000). The funders had no role in study design, data collection and analysis, decision to publish, or preparation of the manuscript.

### Grant Disclosures
The following grant information was disclosed by the authors:
Natural Science Foundation of Shanghai: 21ZR1414000.

### Competing Interests
The authors declare there are no competing interests.

### Author Contributions

- Hu Lv conceived and designed the experiments, performed the experiments, analyzed the data, prepared figures and/or tables, and approved the final draft.
- Jiaojiao Huang conceived and designed the experiments, performed the experiments, analyzed the data, prepared figures and/or tables, and approved the final draft.
- Xin Zhang performed the experiments, prepared figures and/or tables, and approved the final draft.
- Zhiyong He performed the experiments, prepared figures and/or tables, and approved the final draft.
- Jun Zhang conceived and designed the experiments, performed the experiments, analyzed the data, prepared figures and/or tables, authored or reviewed drafts of the article, and approved the final draft.
- Wei Chen conceived and designed the experiments, performed the experiments, analyzed the data, prepared figures and/or tables, authored or reviewed drafts of the article, and approved the final draft.

### Animal Ethics

The following information was supplied relating to ethical approvals (*i.e.*, approving body and any reference numbers):

All animal-related experiments were reviewed and approved by the Fudan University Shanghai Cancer Center Ethics Committee (FUSCC-IACUC-S2022-0346).

### Data Availability

The raw data are available in the Supplemental Files.

### Supplemental Information

Supplemental information for this article can be found online at http://dx.doi.org/10.7717/peerj.16855#supplemental-information.

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
