# Peer review of "Xenon ameliorates chronic post-surgical pain by regulating mitophagy in microglia and rats mediated by PINK1/Parkin pathway"

_PeerJ, doi:10.7717/peerj.16855_

## Round 0.1 · original submission · Major Revisions

Your manuscript has been evaluated by three experts, with comprehensive instructions for improvement - please take care to address all queries in full.

·

Basic reporting

In this manuscript, Lv et al. demonstrated the role of Xenon in relieving Chronic post-surgical pain (CPSP). The authors show that Xe treatment suppresses microglial activation and oxidative stress. Further, Xe promotes PINK1/Parkin-mediated mitophagy. The manuscript is well written and results justify the conclusions. However, there are certain concerns that should be addressed:

1. It would be helpful to include a schematic diagram showing how the SMIR model was constructed.
2. Line 238 "day 2, 4, 6, 8, 10, 12 and 14 (short-term)" should be long-term.
3. Combine Figures 1 and 2.
4. Figure 3. Quantify Iba-1 intensity. The scale bars are hard to see. Also scale bar information in the figure legends.
5. Figure 4. Add scale bar to the images. Include in the text the explanation of why ATG5, Parkin and BECN1 levels were tested.
6. Figure 5A. Quantify the degree of Iba-1/PINK1 co-localization. Put 5B and C as supplementary information. The methods state that two siRNAs were generated. Which siRNA was used for the subsequent experiments? Include this information. For the Mito-Sox and Mito-tracker experiments, include proper siRNA controls (CPSP and siPINK1, CPSP and scramble). Include scale bars for all experiments.
7. Combine Figures 6 and 7
8. Figure 7. Quantify Iba-1 intensity. Include scale bars. Include proper siRNA controls for all the experiments .

Experimental design

1. Figures 6 and 7 lacks proper experimental controls which should be included.
2. In the methods section: What is the composition of the lysis buffer used for western blot?
3. Expand "SD" rats in line 93.
4. The authors state in the discussion that "In the past decades, the protective effect of xenon on the central nervous system has been demonstrated (Fries et al. 2008)." The authors should describe in detail what knowledge gap still exists and how this study fills the gap.

Validity of the findings

1. Lines 373-375- "The current study confirmed that Xe has the effect of inhibiting the promotion of mitophagy and inhibiting oxidative stress in the rat model of CPSP, which may be a potential drug for the treatment of CPSP." The authors are showing that Xe promotes mitophagy. This sentence should be corrected.

2. Lines 285-287- "These results demonstrated that Xe treatment promotes microglia autophagy and reduces mitochondrial dysfunction, thereby reducing oxidative stress in microglia." This conclusion is overreaching since the authors did not test mitochondrial function. This sentence should be toned down.

3. The authors show that CPSP induces mitophagy and this phenotype is further amplified on Xe treatment. The authors need to discuss in detail why Xe is having a protective effect when it worsens the CPSP phenotype (increased mitophagy). Additionally, Xe is suppressing oxidative stress, meaning that it is making the mitochondria healthier. But then, why is there increased mitophagy?

4. Does Xe affect mitochondrial biogenesis? It would be beneficial to test some mitochondrial biogenesis markers (PGCalpha, TFAM etc) via western blot. Also it would be good if the authors could test if Xe alters mitochondrial fission fusion markers (Drp1, Mfn etc). This data will add more depth to the conclusions and make the manuscript more comprehensive.

Additional comments

In the methods section some sentences are grammatically incorrect.

Reviewer 2 ·

Basic reporting

This study investigated the effect of Xenon on chronic post-surgical pain.The activation of microglia in the spinal cord is closely related to the generation, transmission and maintenance of CPSP. The study showed that Xe treatment in CPSP rat model could significantly improve PWMT and TWL. Xe treatment could decrease the expression of Iba1 and MDA/H2O2 levels and the increase of SOD/CAT levels. Mechanistically, Xe promoted microglia mitophagy and inhibited oxidative stress in CPSP rat model. This study suggests that Xe plays a role in ameliorating CPSP by regulating the PINK1/Parkin pathway mediated microglial mitophagy and provide new ideas and targets for the prevention and treatment of CPSP.
Although this study is clinically significant and worthy of publication, there are several major concerns that need to be addressed.
1. The description "day2,4,6,8,10,12,14(short-term)" in line 238 in the first part of the results should be "day2,4,6,8,10,12,14(longterm)". Also, the description of the results in line 240-241 in the section "Behavioral testing showed that PWMT was significantly increased after CPSP, but no statistical difference after 2 h compared with CPSP group (Figure 2A)" is not accurate, please check it again.
2. Figures2 A-D and Fig7D-G are not presented clearly enough, especially the part of statistical difference, is it correct to mark * and #, please check.
3. Most of the IF results lack statistical analyses, such as Fig3E, Fig5A, Fig7A, please provide.
4. The description of Fig6 is missing in the results.

Experimental design

no comment

Validity of the findings

Conclusions are well stated. However, Mitophagy flux analysis must be performed in the presence/absence of later-stage lysosomal acidification inhibitors or lysosomal/autophagosome fusion inhibitors such as chloroquine or Bafilomycin. This is a crucial aspect because
it clearly defines if an increase in mitochondrial/lysosomal colocalization is an indication of increased mitophagy. Also a mitophagy block in the latter stage could show the accumulation of colocalized particles. Therefore, only by blocking lysosomal cargos degradation it is possible to address the effective mitophagy flux.

Additional comments

Please add specific methods of colocalisation of mitophagy in the methodology

·

Basic reporting

The paper reports deciphering the in vitro and in vivo regulatory effects of an anesthetic gas Xenon in chronic post-surgical pain (CPSP) experimental model in rats. Structure of the article and the language use follow professional standards, data are shared in Excel as a supplement. However, Fig. 1 and Fig.6 look nearly similar because the only difference includes the notification about the experimental methods employed at the final step of the protocol on day 14 (abbreviations for 5 vs 2 methods are shown, respectively). To improve the structure, the universal figure looks optimal, with description of methods used in Material and Methods section and/or in the text where appropriate.

Experimental design

The experimental study fits the Aims and Scope of the journal. Methods are described with necessary details. The data followed high ethical standard since the local AICUC approval was received for this particular study. However, translation of the IACUC Approval Notice to English is desirable.

Validity of the findings

Although the study is novel, "there are few studies on the efficacy of xenon on CPSP"(lane 82, no citation for this statement). Additional citation of even few studies may further improve the structure of the paper.
The most meaningful finding of the study is the simultaneous effect of Xenon on both microglial activation, oxidative stress in rats and relieving CPSP induced in experimental animals. Most importantly, for the first time it was shown that the pain and proinflammatory markers were both significantly decreased even 10 days after the last exposure of animals to Xenon using the five-day exposure protocol, one exposure/day. These and other results are based on valid data confirmed by statistical analysis.
The data of the experimental study support the conclusion that multiple Xenon exposures are required for chronic neuropathic pain treatment that links to PINK1/Parkin pathway mediated microglial mitophagy.

Additional comments

Most importantly, the results of the successful experimental treatment of CPSP provide valid support for a future clinical trials. This valuable message deserves to be shared among clinical specialists .

---

## Round 0.2 · Minor Revisions

Reviewer 2 has highlighted two points for clarification, please address these accordingly.

·

Basic reporting

I just noticed a few typos. Other than that, the authors have addressed all my comments. The manuscript is now suitable for publication.

Experimental design

N/A

Validity of the findings

N/A

Additional comments

N/A

Reviewer 2 ·

Basic reporting

The study showed that Xe treatment in CPSP rat model could significantly improve PWMT and TWL. Xe treatment could decrease the expression of Iba1 and MDA/H2O2 levels and the increase of SOD/CAT levels. Mechanistically, Xe promoted microglia mitophagy and inhibited oxidative stress in CPSP rat model. This study suggests that Xe plays a role in ameliorating CPSP by regulating the PINK1/Parkin pathway mediated microglial mitophagy and provide new ideas and targets for the prevention and treatment of CPSP.
Howvere, the revised manusccipt have several minor concerns that need to be further addressed.
1.Please add specific methods of colocalisation of mitophagy in the methodology. (Although the author says it's been described, I didn't find it in the revised manuscript)
2.The title in revised manuscript is not quite right, so I suggest double-checking it.

Experimental design

no commnets

Validity of the findings

Mitophagy flux analysis must be performed in the presence/absence of later-stage lysosomal acidification inhibitors or lysosomal/autophagosome fusion inhibitors such as chloroquine or Bafilomycin. This is a crucial aspect because it clearly defines if an increase in mitochondrial/lysosomal colocalization is an indication of increased mitophagy. Also a mitophagy block in the latter stage could show the accumulation of colocalized particles. Therefore, only by blocking lysosomal cargos degradation it is possible to address the effective mitophagy flux.

---

## Round 0.3 · accepted · Accept

The second round of revisions has adequately addressed the suggestions from the reviewer.